# Finding the Dose for Ceftolozane-Tazobactam in Critically Ill Children with and without Acute Kidney Injury

**DOI:** 10.3390/antibiotics9120887

**Published:** 2020-12-10

**Authors:** Laura Butragueño-Laiseca, Iñaki F. Troconiz, Santiago Grau, Nuria Campillo, Xandra García, Belén Padilla, Sarah N. Fernández, María José Santiago

**Affiliations:** 1Pediatric Intensive Care Unit, Hospital General Universitario Gregorio Marañón, 28009 Madrid, Spain; laura_bl@hotmail.com (L.B.-L.); sarahlafever@gmail.com (S.N.F.); 2Gregorio Marañón Health Research Institute (IISGM), 28007 Madrid, Spain; 3Pediatrics Department, Universidad Complutense de Madrid, 28040 Madrid, Spain; 4Maternal and Child Health and Development Research Network (REDSAMID), Institute of Health Carlos III, 28029 Madrid, Spain; 5Pharmacometrics & Systems Pharmacology Research Unit, Department of Pharmaceutical Technology and Chemistry, School of Pharmacy and Nutrition, University of Navarra, 28027 Pamplona, Spain; itroconiz@unav.es; 6IdiSNA, Navarra Institute for Health Research, 31008 Pamplona, Spain; 7Pharmacy Department, Hospital del Mar, Universitat Autònoma de Barcelona, 08193 Barcelona, Spain; sgrau@hospitaldelmar.cat (S.G.); NCampillo@parcdesalutmar.cat (N.C.); 8Pharmacy Department, Hospital General Universitario Gregorio Marañón, 28009 Madrid, Spain; xandra.garcia@salud.madrid.org; 9Clinical Microbiology Department, Hospital General Universitario Gregorio Marañón, 28009 Madrid, Spain; belenpadilla@icloud.com

**Keywords:** ceftolozane, acute kidney injury, population pharmacokinetics, critically ill children, dose individualization, continuous renal replacement therapy

## Abstract

**Background:** Ceftolozane-tazobactam is a new antibiotic against multidrug-resistant pathogens such as *Pseudomonas aeruginosas*. Ceftolozane-tazobactam dosage is still uncertain in children, especially in those with renal impairment or undergoing continuous renal replacement therapy (CRRT). **Methods:** Evaluation of different ceftolozane-tazobactam dosing regimens in three critically ill children. Ceftolozane pharmacokinetics (PK) were characterized by obtaining the patient’s specific parameters by Bayesian estimation based on a population PK model. The clearance (CL) in patient C undergoing CRRT was estimated using the prefilter, postfilter, and ultrafiltrate concentrations simultaneously. Variables such as blood, dialysate, replacement, and ultrafiltrate flow rates, and hematocrit were integrated in the model. All PK analyses were performed using NONMEM v.7.4. **Results:** Patient A (8 months of age, 8.7 kg) with normal renal function received 40 mg/kg every 6 h: renal clearance (CL_R_) was 0.88 L/h; volume of distribution (Vd) Vd_1_ = 3.45 L, Vd_2_ = 0.942 L; terminal halflife (t_1/2,β_) = 3.51 h, dosing interval area under the drug concentration vs. time curve at steady-state (AUC_τ,SS_) 397.73 mg × h × L^−1^. Patient B (19 months of age, 11 kg) with eGFR of 22 mL/min/1.73 m^2^ received 36 mg/kg every 8 h: CL_R_ = 0.27 L/h; Vd_1_ = 1.13 L; Vd_2_ = 1.36; t_1/2,β_ = 6.62 h; AUC_SS_ 1481.48 mg × h × L^−1^. Patient C (9 months of age, 5.8 kg), with severe renal impairment undergoing CRRT received 30 mg/kg every 8 h: renal replacement therapy clearance (CL_RRT_) 0.39 L/h; Vd_1 =_ 0.74 L; Vd_2=_ 1.17; t _1/2,β =_ 3.51 h; AUC_τ,SS_ 448.72 mg × h × L^−1^. No adverse effects attributable to antibiotic treatment were observed. **Conclusions:** Our results suggest that a dose of 35 mg/kg every 8 h can be appropriate in critically ill septic children with multi-drug resistance *Pseudomonas aeruginosa* infections. A lower dose of 10 mg/kg every 8 h could be considered for children with severe AKI. For patients with CRRT and a high effluent rate, a dose of 30 mg/kg every 8 h can be considered.

## 1. Introduction

Multi-drug resistance (MDR) among Gram-negative bacteria has increased in recent years and is recognized as a major threat worldwide [1]. Infections produced by *Pseudomonas aeruginosa* have become a real concern in hospital-acquired infections, especially in critically ill and immunocompromised patients. New antibiotics are needed to treat infections caused by MDR Gram-negative bacteria. Ceftolozane-tazobactam has been developed to overcome *P. aeruginosa*´s antimicrobial mechanisms of resistance such as changes in porin permeability and upregulation of efflux pumps [2]. Ceftolozane-tazobactam has proven to have a high in vitro activity against the majority of MDR *P. aeruginosa* strains. In vivo, ceftolozane-tazobactam showed positive outcomes in 71% of patients with MDR *P. aeruginosa* infections [3,4]. Pharmacokinetic (PK) studies for new antibiotics are important in critically ill patients because there can be marked changes in PK due to the presence of systemic inflammatory response syndrome, capillary leak syndrome, hypoalbuminemia, or dysregulation in renal blood flow [5].

Moreover, acute kidney injury is a common complication in critically ill children. One in four children admitted to the intensive care unit (ICU) will develop acute kidney injury and 10 to 15% will need continuous renal replacement therapy (CRRT) [6]. Antibiotic dosing in critically ill patients undergoing CRRT is challenging [7,8]. Limited data are available regarding ceftolozane-tazobactam PK in children and adolescents [9,10]. There are no studies in children undergoing CRRT. Therefore, in this analysis, we describe the cases of three critically ill children treated with different ceftolozane dosing regimens based on renal function or the use of CRRT. The aim of this study was to characterize the PK of ceftolozane in these three critically ill pediatric patients and discuss possible dosing regimens for children.

## 2. Methods

The results of this study are part of a broader project intended to describe the PK properties of several antibiotics in pediatric patients with CRRT (European Regional Development Fund (ERDF), ref. RD16/0022/0007). All patients receiving antibiotics and with an intravenous catheter allowing needle-free blood draws in the pediatric intensive care unit were eligible for this prospective study. The study was approved by the Gregorio Marañon Institutional Review Board. Informed consent from the parents was obtained for all the patients included in the study. Those who were treated with ceftolozane-tazobactam are presented in this paper.

### 2.1. Sampling and Analytical Method

One mL blood and urine samples were collected in heparinized tubes before (T0) and 2, 4, 6, and 8 h after starting the infusion (T2, T4, T6, T8). In the patient with CRRT, blood samples from the prefilter and postfilter ports and a sample from the effluent port of the Prismaflex^®^ (Baxter Int.) device were drawn simultaneously. Samples were centrifuged at 2500 rpm for 10 min to yield at least 350 mcL of plasma. Plasma samples were stored at −80 °C until the assay. The samples were analyzed in the Pharmacy Department of the Hospital del Mar, Barcelona, Spain. Ceftolozane concentrations were quantified using a validated high-performance liquid chromatography (HPLC) method.

### 2.2. Data Analysis

Due to the fact that this report includes a very small number of patients with sparse sampling, no attempts were made to develop a population PK model. Patients’ individual PK parameters were obtained mainly by Bayesian estimations supported by an available previous population PK [9] by providing the patient’s characteristics. Analyses were performed with NONMEM software 7.4 version [11] using the option of MAXEVAL = 0 to get the empirical Bayes estimates of the parameters and the first order estimation method to get the corresponding parameters for the hemodialyzed patient as detailed below.

For the patient with CRRT: the concentrations of ceftolozane measured in prefilter (C_Pre_), postfilter (C_Post_), and effluent (C_Efflu_) ports were analyzed simultaneously to estimate the renal replacement clearance (CL_RRT_) based on a novel approach called the integrated dialysis pharmacometric (IDP) model [12].

Detailed information regarding the population PK analysis is provided in the appendix.

### 2.3. Pharmacodynamics

Betalactam antibiotics show time-dependent antibacterial activity, therefore the primary pharmacokinetic/pharmacodynamic target is the time that the unbound plasma drug concentration (C_u_) remains above the minimum inhibitory concentration (MIC) of the infecting pathogen. In the current evaluation, values of C_u_ were obtained as the product between the model predicted plasma concentrations and the unbound fraction in plasma (f_u_). The f_u_ of ceftolozane is reported to be 0.8 [4].

The therapeutic goal for cephalosporins is to achieve values of Cu greater than the MIC for at least 40% of the dosing interval, although 100% may be desirable for optimal outcomes in critically ill patients [1,7,10]. MIC breakpoint for *Pseudomonas aeruginosa* was ≤4 µg/mL according to EUCAST and CLSI guidelines [13].

## 3. Results

### 3.1. Patient Characteristics, Dosing and Sampling

Table 1 lists the demographic characteristics and underlying condition of the three patients involved in the current study. A MDR *P. aeruginosa* susceptible to Ceftolozane (with a MIC ≤4 μg/mL) was isolated from the biological specimens in all three cases.

Table 2 presents the ceftolozane dosing regimen for each patient. The dose was decided by the treating physicians of each patient according to the available data in the scientific literature [14,15] and the patient´s condition.

Patient A was septic, but without acute kidney injury, so a standard dose of 40 mg/kg every 6 h by intravenous infusion over one hour was used [14,15]. Patient B had stage II AKI according to the KDIGO classification [16], so the dose was adjusted to 36 mg/kg every 8 h. Patient C was on CRRT with continuous venovenous hemodiafiltration (CVVHDF) and received a dose of 30 mg/kg every 8 h. CRRT settings in this case were: blood flow 30 mL/minute, substitution flow 160 mL/h, dialysis flow 250 mL/h, extraction 60 mL/h, and total effluent flow 470 mL/h.

Data from 17 concentration–time data points were available for population pharmacokinetic analysis. Measured blood concentrations of ceftolozane are displayed in Table 2. All measurements throughout the dosing interval were above the ceftolozane MIC.

### 3.2. Pharmacokinetics of Ceftolozane in Patients without Continuous Renal Replacement Therapy (CRRT)

Table 3 lists the PK parameters obtained from the analysis together with the relevant PK metrics. As can be observed in Figure 1, those parameters provided a good description of the concentrations of ceftolozane in serum (panels A and B) and in urine (panel C). In the case of patient A, there was an over prediction of the trough levels after the second administration, which was negligible at the end of treatment.

The estimated renal clearance was quite different between the two patients (0.88 vs. 0.27 L/h), which is explained by the differences in the estimated glomerular filtration rate (75 vs. 22 mL/min/1.73 m^2^). The terminal half-life of patient A (3.51 h) was approximately half of the one obtained in patient B (6.62 h) due to the fact that despite CL_Renal_ being 3.26 fold higher, the Vd apparent volume of distribution at steady state (V_SS_) calculated as the sum of Vd_1_ and Vd_2_ was also higher in patient A (4.4 vs. 2.5 L).

According to the model developed by Larson et al. [9], the PK parameter values for patients with the same weight and eGFR (estimated glomerular filtration rate) as the ones in our evaluation were: Patient A [1.02 L (Vd_1_), 1.54 L (Vd_2_), and 0.83 L/h (CLrenal)] and patient B [1.33 L (Vd_1_), 1.73 L (Vd_2_), and 0.37 L/h (CL_Renal_)]. For both patients, the predicted individual PK profiles lay within the 90% predicted intervals computed from the Larson et al. model for individual characteristics equal to those of patients A and B, as shown in Figure 1A,B).

### 3.3. Pharmacokinetics of Ceftolozane in the Hemodialyzed Patient

Table 3 lists the PK parameters obtained during the analysis and the relevant PK metrics for the hemodialyzed patient (patient C). Similarly, as can be observed in Figure 2A), those parameters provided a good description of the three types on concentrations (C_pre_, C_post_, and C_Efflu_).

The estimates of CL_RRT_ and V_SS_ were 0.39 L/h and 1.91 L, respectively. The calculated terminal half-life was 3.51 h. The value of BPR was estimated in 1.28, indicating that concentrations of ceftolozane in blood were 30% higher than those in plasma. Calculations made with the reference model of Larson et al. (2019) for the PK parameter values for a patient with the same weight as patient C were 0.64 L (Vd_1_) and 1.27 L (Vd_2_). The predicted individual PK profiles of patient C lay within the 90% predicted intervals computed from the Larson et al. (2019) model [9], as shown in Figure 2, assuming a value of eGFR of 80 mL/min/1.73 m^2^, indicating an adequate elimination performance of the hemofilter. After the last administered dose, the model predicted a cumulative excreted amount of ceftolozane in urine of 165 mg at the end of the dosing interval (panel C).

Values of extraction ratios (see the appendix for calculation) were 0.82 (at trough), 0.85 (T1), 0.86 (T2), and 0.75 (T8). The values of the Sieving coefficients were: 1 (at trough), 1.1 (T1), 0.99 (T2), and 1.14 (T8).

### 3.4. Pharmacodynamic Target

For dose selection, the first PK criterion acted as a maximum exposure bound, the 95th percentile of adult ceftolozane exposure (AUC_0–8_ = 628 (mg·h/L) and C_max_ = 151 μg/mL) was used [4]. The second PK criterion was a C_min_ = 16 for ceftolozane, and acted as a potential efficacy bound.

The second criterion was met in all three patients as not only were all plasma concentrations throughout the dosing interval above the ceftolozane MIC, but the C_min_ was ≥16.

The first criterion was also accomplished for patients A and C, but patient B exceeded this criterion with an AUC_0–8_ = 1481 (mg·h/L) and a Cmax = 220 mg/L.

In patient B, a dose of 20 mg/kg/8h would have achieved steady state values of C_min_ = 55 mg/L, C_max_ = 143 mg/L and AUC_SS_ = 484 mg·h /L.

Furthermore, a dose of 10 mg/kg/8h would have also achieved steady state values of C_min_ = 23 mg/l, C_max_ = 74 mg/L, and AUC_SS_ = 250 mg.h/L.

## 4. Discussion

Pharmacokinetic studies for new antibiotics are important in critically ill patients because intense systemic inflammatory response syndrome (SIRS) can cause marked changes in pharmacokinetics. The presence of capillary leak syndrome, hypoalbuminemia, or disregulations in renal blood flow contribute to these changes. Subtherapeutic dosing is one of the major contributing factors for antibiotic resistance appearance and mortality. This situation is even more important in pediatric patients as there is no information on a population usually excluded from clinical trials with new antibiotics. Therefore, individualizing the dose of antibiotics according to the patients’ clinical situation is essential to maximize favorable outcomes [5,7,8].

The pharmacokinetics of ceftolozane and tazobactam have been investigated in healthy adult volunteers [17], adults with renal impairment [18], and adult patients with abdominal and complicated urinary infections [19].

In healthy adults, ceftolozane has low protein binding (20%), a half-life of 3.12 h and V_SS_ of 13.5 L. More than 95% of the drug is renally eliminated through glomerular filtration [4,10]. In adults, ceftolozane dosing requires dose adjustment in patients with creatinine clearance ≤50 mL/min [4].

Currently, data regarding ceftolozane-tazobactam PK in children and adolescents are scarce [9,10,20]. A recent phase I clinical study evaluating the PK, safety, and tolerability of single i.v. ceftolozane-tazobactam doses in pediatric patients (birth to 18 years old) with proven/suspected Gram-negative bacterial infections [10] found that exposures calculated using non-compartmental analysis were comparable to those reported for adults. In addition, ceftolozane-tazobactam was well tolerated, and no safety concerns were identified.

However, dose finding PK studies in healthy children populations do not usually result in optimal treatments for critically ill patients [7,21].

The three patients studied in the current evaluation represent the three different scenarios concerning renal function in critically ill children: normal renal function, AKI, and AKI requiring CRRT. In patient A, with a normal renal function and a critical condition, the dose decided was higher than the dose used in Bradley´s study [10] (Table 4).

The terminal half-life and the V_SS_ in our patient were slightly higher than observed in Bradley´s study [10].

This might be explained because critically ill children usually have higher total body and extracellular water than other child populations as a certain amount of fluid overload is commonly present in these patients. This increase in total body and extracellular water can result in an increase of V_SS_ (and consequently half-life) for water soluble drugs such as ceftolozane [14]; in addition, the apparent volume of distribution Vd seems to be slightly higher in young infants and neonates than older children [13] and adults.

With respect to elimination, the estimated renal clearance in patient A is comparable with CL_Renal_ for group number 4 in Bradley’s study (Table 4). The ceftolozane renal clearance seems to be lower in toddlers than in older children [10] and adults [17]. These differences are consistent with age-related physiologic changes as renal function matures through infancy, with clearance values reaching adult values around the second year of life. For instance, the estimated glomerular filtration rate for our first patient, who was eight months old, was 75 mL/min/1.75 m^2^, which is normal for his age [22].

The second case (patient B) had an impaired renal function (stage II AKI according to the KDIGO classification) ([16] with a eGFR 22 mL/min/1.73 m^2^. The impact of differences in creatinine clearance across patients in the degree of exposure to ceftolozane-tazobactam has been previously studied in adults [18]. A 2.5-fold increase in the AUC for ceftolozane was found in subjects with moderate renal impairment. In case of subjects with severe renal impairment, the AUC for ceftolozane increased 4.4-fold. In end-stage renal failure subjects, ceftolozane concentrations declined rapidly following the start of hemodialysis (HD), with approximately 66% reduction in AUC with respect to pre-dialysis. Slight increases in exposure with mild renal impairment do not warrant a dose adjustment; however, subjects with moderate or severe renal impairment (patient B) and those on HD require a decrease in the dose or a change in the frequency of administration [18].

### 4.1. CRRT Clearance

Ceftolozane is a relatively small molecule (molecular weights of 666 Daltons) with a low plasma protein binding rate (20%) [4]. These properties suggest that the drug will be removed by CRRT.

In an ex vivo CRRT experiment [23], no statistically significant differences were observed in the sieving coefficient (1.03 ± 0.11) between hemofilter types or ultrafiltrate flow rates or RRT modality. Increases in effluent flow and dialysis flow directly increases transmembrane clearance. This is important in children because we normally use moderate and high effluent flows (especially when using regional citrate anticoagulation) [24].

We observed extraction ratios (0.82–0.86) and sieving coefficients (0.99–1.14) that were consistent with previous findings for continuous hemofiltration and dialysis [23,25,26] in adults.

The total estimated ceftolozane clearance rate during CVVHDF (0.39 L/h) was about half of that for patient A, which had normal renal function. This has also been described in adults studies [25,26] with values ranging from 7.2 L/h in patients with normal renal function vs. 3.3 L/h in patients with CRRT.

Dosing recommendations for CRRT are not transferable to intermittent hemodialysis due to the differences in pharmacokinetics between both renal replacement strategies [27]. Specific PK studies in children undergoing CRRT as well as further research including larger pediatric population studies are necessary in order to provide more robust dosing recommendations.

### 4.2. Pharmacodynamic Target and Area under Curve (AUC)

All patients achieved a high area under curve (AUC) at steady state. Like other beta-lactams, the PK/pharmacodynamic (PD) target that best correlates with ceftolozane efficacy is the amount of time (as a proportion of the total dosing interval) that the free drug concentration remains above the MIC, expressed as the %fT MIC.

Therefore, a %fT MIC of 30% (of an 8-h dosing interval) for a MIC ≤4 μg/mL can be assumed to be a suitable PK/PD target for ceftolozane, based on existing regulatory PK/PD guidance for the development of antibacterial treatments for potentially life-threatening infections [13].

However, given that susceptibility data are usually not available when ceftolozane is started, and that an initial empirical coverage at a higher target for ceftolozane (100% fT > MIC) could be advantageous in critically ill patients [21].

The trough concentrations seen in patients A and C were between four and 10 times the MIC breakpoint for *P. aeruginosa* (4 μg/mL). Therefore, these concentrations are generally acceptable given that most experts consider trough concentrations above 10 times the MIC to represent a cutoff point for dose reduction of beta-lactam antibiotics (not because of significant toxicity concerns, but for avoidance of unnecessary high exposures) [21]. In addition, keeping the concentration above four to five times the MIC has been shown to maximize the antibacterial effect of beta-lactams [21]. Furthermore, given the poor reproducibility of MIC measurements, it is not uncommon that an isolate considered susceptible at the breakpoint MIC is subsequently found to actually be resistant with retesting, with a MIC higher by up to two dilutions (up to 16 μg/mL) [25].

Although there is no clearly defined toxicity threshold for ceftolozane steady-state concentrations, a high dose of 40 mg/kg every 6 h appears to achieve unnecessarily high steady state ceftolozane concentrations. According to our results, a dose of 35 mg/kg every 8 h could be used in septic patients with infections caused by MDR *Pseudomonas aeruginosa*.

Doses of 10 mg/kg every 8 h could be sufficient in children presenting AKI with a dose reduction similar to that proposed in adult patients with this pathology [4,18].

Only the pharmacokinetic information of one patient undergoing CRRT has been included. In patient C, with CRRT and a high effluent rate, a dose of 30 mg/kg every 8 h could be enough. Given the characteristics of these patients, it would be advisable to perform multicenter studies in critically ill children enrolling enough patients with AKI and CRRT.

## 5. Conclusions

Overall, the current analysis demonstrated the relevance of population PK in clinical practice in special populations and highlights the importance and the urgent need of developing predictive specific models for those critical patients. In addition, it shows the possibility of integrating drug concentrations measured in serum and urine, and pre-, post-filter, and effluent, to get the most of the data, especially in a clinical setting where rich sampling is not always possible.

Our results suggest that a dose of 35 mg/kg every 8 h can be appropriate in critically ill septic children with MDR *Pseudomonas aeruginosa* infections. A lower dose of 10 mg/kg every 8 h could be considered for children with severe AKI. For patients with CRRT and a high effluent rate, a dose of 30 mg/kg every 8 h can be considered.

## 6. Limitations

The main limitation of this paper is the reduced number of patients. Multicenter studies with an adaptable and flexible design including enough patients are necessary to offer more robust dosing recommendations in children.

Only one patient had more than one curve of measurements as we attempted to minimize the amount of overall blood collection from each patient.

Filter size and ultrafiltration flows are important determinants of antibiotic clearance. Their use is not stable or steady in pediatric patients, which is why more studies are necessary to adjust the effect of these factors in the PK of ceftolozane.

## 7. Addendum to the Material and Methods Section

### 7.1. Population Pharmacokinetics (PK) Modeling Analysis

#### 7.1.1. Patients without Hemodialysis

The PK of ceftolozane has been previously characterized in a population of 452 patients including 31 children between seven days and 18 years of age with complicated urinary tract infections (Bradley et al., 2018) [10] by means of a population PK analysis (Larson et al., 2019) [9].

Ceftolozane plasma and urine concentrations over time were described using a two compartment model where apparent volumes of distribution and total plasma clearance were allometric scaled by body weight relative to a 70 kg subject, and total plasma clearance was related to the glomerular filtration rate calculated by either the modification of diet in renal clearance for adult patients, or the modified Schwartz formula [28] in the case of children as follows:(1)$$V_{1.i}=10.64\times {\left(\frac{WGT_{i}}{70}\right)}^{1.124}\times e^{\eta_{V1,i}}$$
(2)$$V_{2.i}=4.227\times {\left(\frac{WGT_{i}}{70}\right)}^{0.484}\times e^{\eta_{V2,i}}$$
(3)$$CL_{D.i}=2.545\times e^{\eta_{CLD,i}}$$
(4)$$CL_{Renal,i}=5.88\times {\left(\frac{WGT_{i}}{70}\right)}^{0.764}\times {\left(\frac{eGFR_{i}}{eGFR_{0}}\right)}^{0.7036}\times e^{\eta_{CL,i}}$$
where *V*_1_ and *V*_2_ are the apparent volumes of distribution of the central and peripheral compartments, respectively; *CL_D_* and *CL_Renal_* represent the distribution and total elimination clearances, respectively; *WGT* corresponds to body weight; and *eGFR* is the estimated glomerular filtration rate in the *i*th patient. *eGFR*_0_ is the reference filtration rate set at 173 mL/min/1.73 m^2^.

All the coefficients shown in Equations (1)–(4) correspond to typical parameters estimated in a previous population PK analysis (Larson et al., 2019). The discrepancy between the subject’s specific and typical PK parameters are given by *η*. For each parameter in the model, the ηs forms a random variable symmetrically distributed around 0 with variance ω^2^, which provides a quantitative description of the inter-individual variability (IIV). An exponential model for IIV was used, preventing negative values for individual parameters.

The values of ω^2^ reported by Larson et al. (2019) and used in the current evaluation to obtain the Bayesian estimates of the PK parameters of the patients were 0.12, 0.196, and 2.667 × 10^−2^ for *CL*, *V*_1_, and *V*_2_, respectively, IIV was not included for *CL_D_*. A 7.624 × 10^−2^ value for the covariance between ω^2^_CL_ and ω^2^_V1_ was also reported.

The rate of change of ceftolozane amounts between the different compartments of the model is given by the following set of ordinary differential equations:(5)$$\frac{dA_{1}}{dt}=\frac{CL_{D}}{V_{2}}\times A_{2}-\frac{\left(CL_{D}+CL_{Renal}\right)}{V_{1}}\times A_{1}$$
(6)$$\frac{dA_{2}}{dt}=\frac{CL_{D}}{V_{1}}\times A_{1}-\frac{CL_{D}}{V_{2}}\times A_{2}$$
(7)$$\frac{dA_{u}}{dt}=\frac{CL_{Renal}}{V_{1}}\times A_{1}$$
where *A*_1_, *A*_2_, and *A_u_* represent the amounts of ceftolozane in the central compartment (including plasma), peripheral compartment, and urine, respectively. Predicted plasma and urine concentrations were obtained as *A*_1_/*V*_1_ and *A_u_*/*U_Vol_*, respectively, where *U_Vol_* is the measured volume of urine excreted in each urine recovery interval.

Residual variability considered as the difference between observed and predicted observed concentrations was modeled using an error model with an additive (independent from the value of the concentration) and proportional to the concentration components with estimates of the variance equal to 8.78 × 10^−3^, and 2.37 × 10^−2^, respectively (Larson et al., 2019). In the current evaluation, these values were also applied to the concentrations of ceftolozane in urine.

#### 7.1.2. Patient with CRRT

The concentrations of ceftolozane measured in prefilter (C_Pre_), postfilter (C_Post_), and effluent (C_Efflu_) were analyzed simultaneously based on a novel approach called the integrated dialysis pharmacometric (IDP) model (Broeker et al., 2020) [12].

In our analysis, the distribution parameters (*V*_1_, *V*_2_, and *CL_D_*) were obtained as described above for the two non-hemodialyzed patients, and the same error model with its corresponding variances were used to account for the residual variability in the three types of concentrations.

*CL_Renal_* was absent in this patient and the elimination process was represented by the patient’s individual renal replacement clearance (*CL_RRT_*), which was one of the parameters estimated in the analysis. *C_Pre_*, *C_post_*, and *C_Efllu_* were integrated in the modeling framework as follows:
(i)*C_Pre_* was considered as the predicted plasma ceftolozane concentrations obtained as *A*_1_/*V*_1_ using the set of Equations (1)–(7), in which the entire Equation (4) was substituted by the parameter *CL_RRT_*.(ii)*C_Post_* and *C_Efflu_* were expressed as a function of *C_Pre_* and *CL_RRT_* as indicated below:(8)$$C_{Post}=C_{Pre}\times \left(1-\frac{CL_{RRT}}{\varphi_{Pl,corr}}\right)$$
(9)$$C_{Efflu}=\frac{CL_{RRT}\times C_{Pre}}{\varphi_{Effluent}}$$
where $\varphi_{Pl,corr}$ and $\varphi_{Effluent}$ are the corrected plasma and total effluent flows, respectively. The value of $\varphi_{Pl,corr}$ was calculated as $\varphi_{Pl,corr}=C_{Pre}\times BPR$, with BPR the blood to plasma concentration ratio, and an additional parameter to be estimated from the model. The total effluent was measured during the study (470 mL/h).

#### 7.1.3. Evaluation of Modeling Results

Pharmacokinetic profiles predicted for drug concentrations in plasma postfilter and in the effluent were compared with the measured values to judge the model and parameter description ability. Urine measured and predicted values were also confronted to evaluate model performance.

In order to compare the PK of ceftolozane in the current set of patients with the patient population studied by Larson et al., 2019, one thousand PK profiles of ceftolozane were generated for each of the three patients using the same demographics (WGT and GFR). In the case of hemodialyzed patients, the lower limit of the GFR according to age group was used (≥80 mL/min/1.73) (Bradley et al., 2018). The median and the 90% prediction intervals were computed and represented graphically together with the patient’s model predictions.

#### 7.1.4. Derived Pharmacokinetic Parameters and Metrics

The terminal half-life (*t*_1/2,*β*_) and the area under the plasma concentration vs. time curve during the dosing interval (*τ*) at steady state (*AUC_τ_*_,*SS*_) were calculated from the PK parameters as follows:$$\beta =\frac{1}{2}\times \left[K_{12}+K_{21}+K_{10}-\sqrt{{\left(K_{12}+K_{21}+K_{10}\right)}^{2}-4\times K_{21}\times K_{10}}\right]$$
$$t_{1/2,\beta }=\frac{\ln\left(2\right)}{\beta }$$
$$AUC_{\tau ,SS}=\frac{Dose}{CL\;\left(or\;CL_{RRT}\right)}$$
where *K*_10_, *K*_12_, *K*_21_ are the first order rate constants of elimination from the central compartment, and the distribution between the central and peripheral compartments, respectively, and were calculated as *K*_10_ = *CL* (*or CL_RRT_*)/*V*_1_, *K*_12_ = *CL_D_*/*V*_1_, *K*_21_ = *CL_D_*/*V*_2_.

#### 7.1.5. CRRT Parameters

The extraction ratio (ER and sieving coefficient (SC) were determined based on observed concentrations using the following equations:$$Extraction\;ratio\;=\;\frac{Concentration\;in\;postfilter\;blood\;sample}{Concentration\;in\;prefilter\;blood\;sample}$$
$$Sieving\;coefficient=\;\frac{effluent\;drug\;concentration}{\left(prefilter\;concentration-postfilter\;concentration\right)/2}$$

## Figures and Tables

**Figure 1 antibiotics-09-00887-f001:**
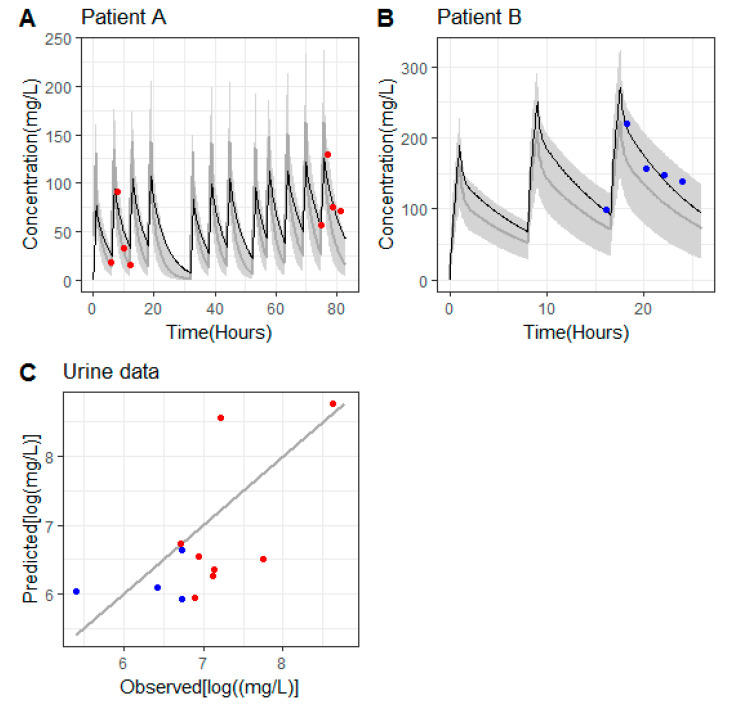
(**A**,**B**) Patients’ model predicted pharmacokinetic profiles of ceftolozane in serum (solid lines in black), measured serum concentrations of ceftolozane (colored circles). Grey areas cover the 90% prediction intervals calculated from 1000 profiles simulated according to the Larson et al. (2019) model [9] based on the weight and eGFR values reported for patients A and B. The solid lines in grey represent the median of the simulated profiles. (**C**) Predicted vs. observed urine concentrations corresponding to patient A (red symbols) and patient B (blue symbols). Solid line is the line of identity.

**Figure 2 antibiotics-09-00887-f002:**
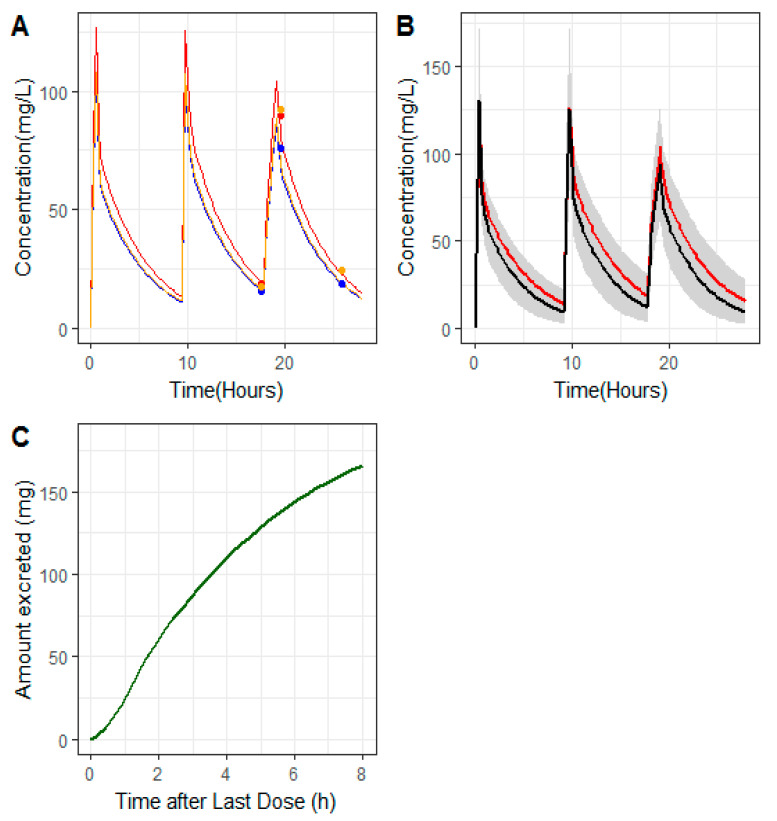
(**A**) Patient C’s model predicted pharmacokinetic profiles (solid lines) of ceftolozane pre-filter (red), post-filter (blue), and in the effluent (orange); solid circles correspond to measured concentrations of ceftolozane in pre-filter (red), post-filter (blue), and effluent (orange). (**B**) Grey areas cover the 90% prediction intervals calculated from 1000 profiles simulated according to the Larson et al. (2019) model [9] based on the weight reported for patient C and assuming an eGFR of 80 mL/min/1.73 m^2^. Red line (patient C’s model predicted profiles), black line (median of the 1000 simulated profiles). (**C**) Predicted cumulative amount excreted through the hemofilter vs. time after the last administered dose.

**Table 1 antibiotics-09-00887-t001:** Demographic characteristics and diagnosis.

Patient	Age (Months)	Weight (kg)	Diagnosis	Infection
A	8	8.7	Hypoplastic left heart syndrome (Norwood stage)	Sepsis and pneumonia due to MDR *P. aeruginosa*
B	19	11	Hypoplastic left heart syndrome (Glenn stage)	Pneumonia due to MDR *P. aeruginosa*
C	9	5.8	Heart transplant	Pneumonia due to MDR *P. aeruginosa* and Bacteremia due to ESBL *Escherichia Coli*

MDR: Multi-drug resistance; ESBL: Extended-spectrum beta-lactamases.

**Table 2 antibiotics-09-00887-t002:** Renal function, dosing, and measured blood concentrations.

Patient	Renal Function (eGFR)	Ceftolozane Dosing Regimen	Dose Number Measured	Ceftolozane Concentrations (g/mL)					
				T0	T1	T2	T4	T6	T8
**A**	75 mL/min/1.73 m^2^	40 mg/kg q6h	2	^a^ 17.8		91.5	33.4	16.1	
**A** **(2nd profile)**	75 mL/min/1.73 m^2^	40 mg/kg q6h	12	^a^ 57.3		129.3	74.9	71.2	
**B**	Acute kidney injury (22 mL/min/1.73 m^2^)	36 mg/kg q8h	3	^a^ 99.1		220.2	156.1	148.2	138.1
**C**	Anuria. CRRT.	30 mg/kg q8h	3	^b^ 18.8	89.5	72.5			24.1
				^c^ 15.5	75.8	62.0			18.1
				^d^ 17.5	92.3	66.5			24.4

^a, b, c, d^ Ceftolozane concentrations measured in the blood, prefilter, postfilter and effluent, respectively.

**Table 3 antibiotics-09-00887-t003:** Pharmacokinetic parameter estimates and derived metrics.

	Non-Hemodialyzed		Hemodialyzed
Parameter	Patient A	Patient B	Patient C
CL_Renal_ (or CL_RRT_) (L/h)	0.88	0.27	0.39
Vd_1_ (L)	3.45	1.13	0.74
Vd_2_ (L)	0.942	1.36	1.17
CL_D_ (L/h) ^1^	2.54	2.54	2.54
BPR (unitless) ^2^	-	-	1.28
t_1/2,_ (h)	3.51	6.62	3.51
AUC_τ,SSC_ (mg × h × L^−1^) ^3^	397.73	1481.48	448.72

CL_Renal_: renal clearances; CL_RRT_: renal replacement therapy clearance; Vd_1_: volume of distribution of the central compartment; Vd_2_: volume of distribution of the peripheral compartment; CL_D_: intercompartmental clearance; BPR: blood to plasma concentration ratio; t_1/2,β_: terminal half-life; AUC_τ,SS_: dosing interval area under the drug concentration vs. time curve at steady-state. ^1^ Estimated without inter-individual variability in the original publication. ^2^ Estimated just for the hemodialyzed patient. ^3^ Calculated at doses of 350, 400, and 175 mg for patient A, B, and C, respectively.

**Table 4 antibiotics-09-00887-t004:** Comparison between patient A and Bradley´s Group #4.

	AUC_06,SS_microg/h/mL Median (Range)	t_1/2_ (h)	V_d_ (L/kg)	CL_Renal_ (L/kg/h)
Patient A (dose: 40 mg/kg)	397.73	3.51	0.505	0.101
Bradley Group 4 (dose: 30 mg/kg) Mean (CV%) (SD lower-upper limits)	202 (158–259)	1.63 (69%) (0.51–2.75)	0.34 (21.1%) (0.27–0.41)	0.149 (43.2%) (0.085–0.213)

AUC: area under curve at steady-state; t_1/2,β_: terminal half-life; Vd volume of distribution: CL_Renal_ renal clearance; CV: coefficient of variance; SD: standard deviation.

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
