# Peer review of "Finding the Dose for Ceftolozane-Tazobactam in Critically Ill Children with and without Acute Kidney Injury"

_antibiotics, 2020, doi:10.3390/antibiotics9120887_

Round 1
Reviewer 1 Report
1) The topic is highly relevant.
2) This is one of the rare cases where PopPK-analysis is applied in a really usefull way (the way it meant to be used).
HOWEVER
The "manuscript" is merely a draft version of a submittable manuscript. It seems to lack any proof-reading by the authors/co-autors.
Usually I reject such draft-versions at once, but in this case not, because in my opinion the topic is important and the authors did a "good thing"
However, the manuscript needs a major revision to be in a version suited to be reviewed.
Please proof-read the manuscript (not only the author, but also the co-authors) and correct it thoroughly!
The "editorial"/language-wise errors are too numerous to list. There are a lot of missing spaces, a lot of typos, a lot of wrong words (throught instead of throughout, through instead of trough, mayor instead of major, ....), some unexplained abbreviations, some wrong abbreviations (MDR --> MRA ?). There are a lot of un-cleaned comments within the manuscript, a lot of crossed-out words,... References are missing sometimes. Some paragraphs are appear twice (in results and in discussion), and so on....
Please prepare a finished version of the manuscript and we can proceed.
Beside the more formal matters listed above please make sure to keep a clear structure: what are the main RESULTS, what is discussion (not to many repeats in results and discussion part) and so on.
Some further specific comments:
- methods: Please mention briefly how the free concentrations have been determined? (ultrafiltration? which UF device?)
- Tab2: some units are missing, e.g. concentration units. (Each figure and table should be fully comprehensible, after reading the legends only).
- [Tab2 vs Fig1: the concentrations of the "second curve" are not really coming from a q8h dosing but from a 6h dosing if I get it right, aren't they?] The second curve is day4? what was the dose regime on day 3?
- Fig1C: Basis of the log-scale is pobably 2, right?
- on page 6 there is a spanish comment: "esta frase no la entiendo bien" ... I assent.
- first paragraph of the discussion seems a little bit out of place. Sure SIRS plays also a role in pediatric patients, but it seems a little bit out of place here.
- the first two paragraphs on page 8 (discussion) are EXACT copies of two paragraphs in the RESULTS part. Please clarify!
- line 280: The concentrations seen in Pat A and C are all between 16-40 mg/L. Really?
Author Response
RESPONSES TO REVIEWERS
First of all, thank you very much for your corrections.
REVISOR 1
1) The topic is highly relevant.
2) This is one of the rare cases where PopPK-analysis is applied in a really usefull way (the way it meant to be used).
HOWEVER
The "manuscript" is merely a draft version of a submittable manuscript. It seems to lack any proof-reading by the authors/co-autors.Usually I reject such draft-versions at once, but in this case not, because in my opinion the topic is important and the authors did a "good thing"
However, the manuscript needs a major revision to be in a version suited to be reviewed.
Please proof-read the manuscript (not only the author, but also the co-authors) and correct it thoroughly!
According to the reviewer recommendations, the paper has been send to all co-authors for a new revision. Results and Discussion: as they have many changes, these have not been highlighted.
The "editorial"/language-wise errors are too numerous to list. There are a lot of missing spaces, a lot of typos, a lot of wrong words (throught instead of throughout, through instead of trough, mayor instead of major, ....), some unexplained abbreviations, some wrong abbreviations (MDR --> MRA ?). There are a lot of un-cleaned comments within the manuscript, a lot of crossed-out words,... References are missing sometimes. Some paragraphs are appear twice (in results and in discussion), and so on....
Please prepare a finished version of the manuscript and we can proceed.
Beside the more formal matters listed above please make sure to keep a clear structure: what are the main RESULTS, what is discussion (not to many repeats in results and discussion part) and so on.
SOME FURTHER SPECIFIC COMMENTS:
methods: Please mention briefly how the free concentrations have been determined (ultrafiltration? which UF device?)
Answer:
We measure total plasma concentrations in prefilter and postfilters samples. The binding of ceftolozane to human plasma proteins is low (approximately 16% to 21%) (Ceftolozane prescribing information).[4]
Effluent samples have no proteins (they don´t pass throught the filter). We measure free fraction in effluent.
- Tab2: some units are missing, e.g. concentration units. (Each figure and table should be fully comprehensible, after reading the legends only).
Answer: we completed the missing units and legends.
- [Tab2 vs Fig1: the concentrations of the "second curve" are not really coming from a q8h dosing but from a 6h dosing if I get it right, aren't they?] The second curve is day4? what was the dose regime on day 3?
Sorry, that was a mistake, the dose and interval was the same during the treatment. That was corrected in the manuscript.
Fig1C: Basis of the log-scale is pobably 2, right?
The reason for using the log scale in both axes is for clarity in the plot. Ranges of urine concentrations vary greatly within and between the patients, and therefore in natural scale several points appear to collapse.
first paragraph of the discussion seems a little bit out of place. Sure SIRS plays also a role in pediatric patients, but it seems a little bit out of place here.
This was erased and resume.
the first two paragraphs on page 8 (discussion) are EXACT copies of two paragraphs in the RESULTS part. Please clarify!
According to the reviewer recommendations, the discussion has been changed.
line 280: The concentrations seen in Pat A and C are all between 16-40 mg/L. Really?
The sentence was changed to: “The trough concentrations seen in patient A and C are between 4 and 10 times the MIC breakpoint for P. aeruginosa (4 mg/liter)”.
REVISOR 2
Overall – nice report on a small population of children receiving ceftolozane-tazobactam. Overall, the data is valuable information to appropriately dose this newer antimicrobial in this patient population. The paper though should be revised for better flow and organization.
Intro Line 49 – Change to “Ceftolozane-tazobactam has been developed..”
Done
Methods –
Overall - Reporting of patient characteristics and doses received by patients should be reported as part of results
According to the reviewer recommendations this data was change to results. Line 84 - refer to patients in this paragraph as "patient A", etc to better align with tables.
Done.
Line 66 - Indicate “pediatric intensive care unit”; it’s implied elsewhere but should be clearly stated here.
Done.
Line 73 – “MRA P. aeruginosa” – is this supposed to be MDR?
Yes, we changed it to MDR
Line 112 – Pharmacodynamic discussion better moved to the discussion at the end of the paper
Line 126 “trough” not “through”
Done
Discussion - Limitations should be part of the discussion before you draw conclusions
According to the reviewer recommendations , we have changed the order of the sections.
Reviewer 2 Report
Overall – nice report on a small population of children receiving ceftolozane-tazobactam. Overall, the data is valuable information to appropriately dose this newer antimicrobial in this patient population. The paper though should be revised for better flow and organization.
Intro
Line 49 – Change to “Ceftolozane-tazobactam has been developed..”
Methods –
Overall - Reporting of patient characteristics and doses received by patients should be reported as part of results
Line 84 - refer to patients in this paragraph as "patient A", etc to better align with tables.
Line 66 - Indicate “pediatric intensive care unit”; it’s implied elsewhere but should be clearly stated here.
Line 73 – “MRA P. aeruginosa” – is this supposed to be MDR?
Line 112 – Pharmacodynamic discussion better moved to the discussion at the end of the paper
Line 126 “trough” not “through”
Discussion -
Limitations should be part of the discussion before you draw conclusions
Author Response
RESPONSES TO REVIEWERS
First of all, thank you very much for your corrections.
REVIEWER 2
Overall – nice report on a small population of children receiving ceftolozane-tazobactam. Overall, the data is valuable information to appropriately dose this newer antimicrobial in this patient population. The paper though should be revised for better flow and organization.
According to the reviewer recommendations, the paper has been send to all co-authors for a new revision. Results and Discussion: as they have many changes, these have not been highlighted.
Intro Line 49 – Change to “Ceftolozane-tazobactam has been developed..”
Done
Methods –
Overall - Reporting of patient characteristics and doses received by patients should be reported as part of results
According to the reviewer recommendations this data was change to results. Line 84 - refer to patients in this paragraph as "patient A", etc to better align with tables.
Done.
Line 66 - Indicate “pediatric intensive care unit”; it’s implied elsewhere but should be clearly stated here.
Done.
Line 73 – “MRA P. aeruginosa” – is this supposed to be MDR?
Yes, we changed it to MDR
Line 112 – Pharmacodynamic discussion better moved to the discussion at the end of the paper
Line 126 “trough” not “through”
Done
Discussion - Limitations should be part of the discussion before you draw conclusions
According to the reviewer recommendations , we have changed the order of the sections.
Round 2
Reviewer 1 Report
Content-wise I already liked the first version. Now, I am also satisfied by the presentation. Much clearer now. Good work.
Minor:
line 194f still sounds a little bit quirky:
"For dose selection, the first PK criterion, acted as a maximum exposure bound, the 95 th percentile of adult ceftolozane exposure (AUC 0–8 = 628 (mg·h/L) and Cmax=151 g/mL) was used [4]."
perhaps something like that?:
"For dose selection, as first PK criterion the 95 th percentile of adult ceftolozane exposure (AUC 0–8 = 628 (mg·h/L) and Cmax=151 g/mL) acted as a maximum exposure bound [4]."